# Analysis of Style-Shifting on Social Media: Using Neural Language Model Conditioned by Social Meanings

**Seiya Kawano[1], Shota Kanezaki[2,1], Angel García Contreras[1],**
**Akishige Yuguchi[3,1], Marie Katsurai[2] and Koichiro Yoshino[1]**

[1]Guardian Robot Project (GRP), RIKEN, Kyoto, Japan
[2]Graduate School of Science and Engineering,  Doshisha University, Kyoto, Japan
[3]Tokyo University of Science

{seiya.kawano,angel.garciacontreras,akishige.yuguchi,koichiro.yoshino}@riken.jp

{kanezaki21, katsurai}@mm.doshisha.co.jp

## Abstract

In this paper, we propose a novel framework for evaluating style-shifting in social media conversations. Our proposed framework captures changes in an individual's conversational style based on surprisals predicted by a personalized neural language model for individuals. Our personalized language model integrates not only the linguistic contents of conversations but also non-linguistic factors, such as social meanings, including group membership, personal attributes, and individual beliefs. We incorporate these factors directly or implicitly into our model, leveraging large, pre-trained language models and feature vectors derived from a relationship graph on social media. Compared to existing models, our personalized language model demonstrated superior performance in predicting an individual's language in a test set. Furthermore, an analysis of style-shifting utilizing our proposed metric based on our personalized neural language model reveals a correlation between our metric and various conversation factors as well as human evaluation of style-shifting.

## 1 Introduction

Style-shifting refers to such changes in a speaker's language style in a conversation as speech acts (Mizukami et al., 2016), vocabulary (Brennan and Clark, 1996), syntax (Reitter and Moore, 2007), prosody (Natale, 1975; Ward and Litman, 2007) according to the context of the conversation, the speaker's background, and the relationship with the interlocutor. For example, formal language is used in business situations; informal language is used in private situations. Style-shifting can also happen within a conversation to show respect or intimacy to fit the situation (Giles et al., 1987; Giles and Ogay, 2007). Such changes in speech style also depend on the presence of people who may be directly or indirectly listening to a conversation, such as on social media (Bell, 1984; Androutsopoulos, 2014; Birnie-Smith, 2016).

Style-shifting is often associated with speech accommodation theory, which describes how speakers adjust their language use to converge or diverge from that of their interlocutors (Giles et al., 1987; Giles and Ogay, 2007). Convergence occurs when speakers adopt a similar style to their partner, aiming to foster understanding, build rapport, and establish a sense of affiliation. Divergence occurs when deviating from the interlocutor's language style to accentuate differences, to assert individuality, or emphasize social identity. Effective style-shifting helps speakers adjust their psychological distance and improve their communication intentions, leading to smoother conversations (Ishihara and Takamiya, 2019). Measuring style-shifting phenomena is important to advance our understanding of conversation dynamics.

Research on stylistic variation has underscored the intrinsic properties of conversational styles and their interplay with various non-linguistic factors (Kruijff-Korbayová et al., 2008; Kang and Hovy, 2021; Basile et al., 2019; Flekova et al., 2016). Furthermore, many studies have quantified the changes in a speaker's language style in a conversation based on the similarity in word choice between utterances (Nenkova et al., 2008; Danescu-Niculescu-Mizil et al., 2011; Ireland et al., 2011; Nasir et al., 2019; Kawano et al., 2020; Brugnoli et al., 2019). However, despite these insights, such word-based methods still face challenges in capturing nuanced changes in style that reflect various social meanings. The term "social meaning" represents a constellation of traits that linguistic forms convey about the social identity of their users, such as group membership, personal attributes, and individual beliefs (Jan-Petter and Gumperz, 2020; Hall-Lew et al., 2021). To evaluate style-shifting effectively, it is essential to go beyond merely observing words used in the

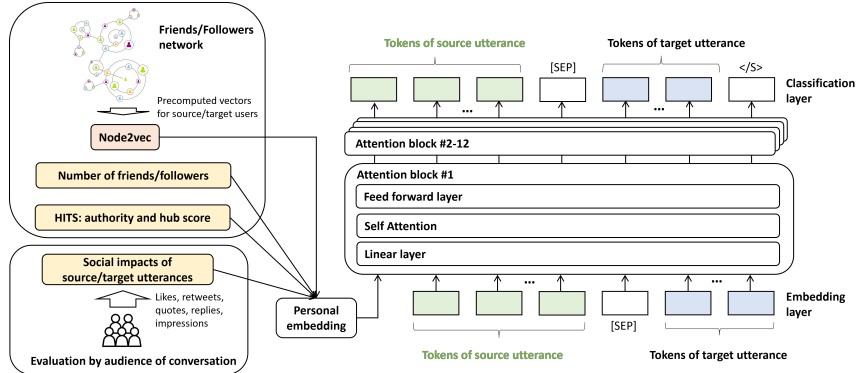

Figure 1: The architecture of our personalized neural language model based on the transformer language model: The green blocks denote linguistic factors, an orange block denotes implicit social meanings and their relationships associated with each user, and yellow blocks reflect the potential social impacts of users and their utterances.

conversation; we must also take into account the content of an individual's usual conversations and the social meanings. In other words, we need a new framework to consider the individual's relative viewpoint and account for non-linguistic factors.

To tackle these problems, we propose focusing on the *surprisal* experienced by a neural language model personalized for an individual. The magnitude of surprisal in the personalized language model is inversely proportional to the consistency between the utterances they are confronted with and the standard language style they support. In other words, the smaller surprisal the more consistent the utterance is with the style of their language; and a larger surprisal indicates that the utterance deviates from their standard language style. The magnitude of surprisal also depends on the incorporated factors of individuals. Comparing surprisal under various conditions could not only assess the level of style-shifting but could also reveal valuable insights into the complex interplay of various factors that influence that change. Furthermore, neural language models are capable of taking into account long-term dependencies between words, deeper meanings, and structures in natural language (Vinyals and Le, 2015; Vaswani et al., 2017; Radford et al., 2018). It can also incorporate non-linguistic factors, such as social meanings, in addition to linguistic factors as conditions for personalizing neural language models for individuals (Li et al., 2016).

Our key contributions are as follows: 1) We introduce a novel metric for evaluating style-shifting, based on surprisal predicted by a personalized neural language model. 2) We develop

a unique personalized language model that incorporates both linguistic and non-linguistic factors. 3) Our personalized language model outperforms existing models in predicting an individual's language use in new conversations, highlighting the effectiveness of incorporating non-linguistic factors into language modeling. 4) We conduct a comprehensive analysis of style-shifting, revealing correlations between our metric and various linguistic and non-linguistic conversation factors as well as human evaluation of style-shifting.

## 2 Neural Language Modeling Conditioned by Social Meanings

In this section, we describe our approach to developing a personalized neural language model integrating both linguistic and non-linguistic factors.

### 2.1 Model Overview

Figure 1 presents an overview of our personalized neural language model. This model generates reply pairs $(S, T)$ from different users ($u_S$ and $u_T$) on social media. The objective is to evaluate the surprisal (or likelihood) of the target utterance $T$ provided by $u_T$, given the source utterance $S$ from $u_S$. Differing from traditional language models for conversation generation, our model extends its consideration beyond mere linguistic factors such as the previous utterance $S$.

A distinctive feature of our model is the introduction of personalized user embeddings as language model conditions. These embeddings are inspired by few-shot learning for personalized dialogue systems and are based on the concept of *homophily*: a theory in sociology that posits individuals with similar social connections tend to

demonstrate similar linguistic behaviors (Laniado et al., 2012; Thelwall, 2010; Chee, 2010). Despite having limited user-specific conversation data, our model is capable of estimating an individual's language use trends. It accomplishes this by leveraging past conversations from similar users and the extensive language knowledge encoded in the pre-trained model (Radford et al., 2018).

Furthermore, our personalized embedding is designed to integrate non-linguistic factors such as the social meanings of users. Here, social meanings span various attributes, including the implicit social meanings and social relationships among individuals, and the social impacts of individuals and their utterances determined by the number of friends/followers and authority and hub scores. Each element offers unique insights into a user's social interactions, thus enriching our understanding of their linguistic behavior. The social impact of a user's utterances, determined through metrics such as likes, replies, and retweets, also significantly influences their language style. By incorporating these factors into personalized embedding, our model effectively captures the unique language styles of individuals.

## 2.2 Language Model Training

In training our personalized neural language model, we consider not only linguistic factors, represented as reply pairs $(S, T)$ from different users ($u_S$ and $u_T$), but also account for non-linguistic factors associated with each user and their utterances. These factors include implicit social meanings and social relationships among users, defined as $v_{u_S \to u_T} = [v_{u_S}; v_{u_T}]$, and the potential social impacts of users and their utterances, which are determined by the number of friends/followers as well as authority and hub scores, defined as $v_I$. Thus, the objective of this model is to minimize the negative log-likelihood of the target utterance given the conditions, as follows:

$$\mathcal{L} = -\log p(T|S, v_{u_S \to u_T}, v_I) \qquad (1)$$

$\mathcal{L}$ is also interpreted as the surprisal of the target utterance $T$, according to information theory. In actual training, $S$ and $T$ are concatenated using a special symbol [SEP], and the model is trained to generate such concatenated text.

We adapt the task of modeling individual language use on social media by fine-tuning a pre-trained model using reply pairs and their associated non-linguistic factors. We employ a transformer language model based on GPT-2 (Radford et al., 2018), which has been pre-trained on a massive corpus of text data. To integrate the non-linguistic factors associated with reply pairs into our language model, we replace the embedding corresponding to the first token of the transformer language model input with a trainable, linearly transformed vector called a personalized embedding.

## 2.3 Personalized Embedding

We introduce two non-linguistic factors to build a personalized embedding: the implicit social relationship between users, and the social impacts of users and their utterances. Both factors are vectorized as $v_{u_S \to u_T} = [v_{u_S}; v_{u_T}]$ and $v_I$, and integrated into the personalized embedding:

**Implicit social relationships**   We calculate the implicit social relationship vector $v_{u_S \to u_T} = [v_{u_S}; v_{u_T}]$ by inputting the friend-follower network of users into the node2vec algorithm (Grover and Leskovec, 2016). The node2vec vectors are pre-computed using the whole network, and the vectors $v_{u_S}$ and $v_{u_T}$ corresponding to $u_S$ and $u_T$ are acquired from the node2vec embedding. We expected this method to encode the structure of the social connections of users, leading to an understanding of their implicit social relationships.

**Social impacts of users and utterances**   We incorporate several factors to calculate the social impacts vector $v_I$, including:

- **Number of friends and followers:** We consider the breadth of social connections of users, as signified by their number of friends and followers. Those with wider social connections could employ a different language style in conversation, reflecting the number of their audiences.
- **Authority and hub scores:** We use the HITS algorithm (Kleinberg, 1999) to evaluate influences of users within their networks. The algorithm provides two scores for each user: the *authority score*, which estimates the value of a user's content based on the number of inbound links from high-hub users, and the *hub score*, which is based on the number of outbound links to high-authority users. A user with a high authority score are recognized by others who follow many influential

users, suggesting that their content is valuable. A user with a high hub score follows many users with high authority scores, indicating their broad awareness of who produces valuable content.

- **Engagement of utterances:** We consider the direct social impacts of utterances through various engagement metrics. These include the number of likes, replies, retweets, and quotes an utterance has received from the community. Users whose posts generate high engagement may exhibit different communication styles compared to those whose posts receive less attention.

After computing vectors $v_{u_S \to u_T}$ and $v_I$, we apply a trainable linear transformation to generate a personalized embedding:

$$v_{personalized} = W[v_{S \to T}; v_I] + b. \quad (2)$$

Here $W$ is the weight matrix, and $b$ is the bias vector, both of which are learned during the model training process. This personalized embedding captures the unique language styles of individuals, making it a powerful tool for understanding language use in social media conversations. By integrating these factors into a comprehensive user embedding, our model more effectively captures the unique language styles of individuals, leading to a more accurate understanding of the dynamics of language use in conversations.

## 3 Evaluation Metric of Style-Shifting

In this section, we describe the metrics used to evaluate style-shifting in our personalized neural language model. Our style-shifting evaluation metrics are based on the surprisal, which is the negative log-likelihood of target utterance calculated from our personalized neural language model conditioned by social meanings. In general, the magnitude of surprisal in an individual's personalized language model is inversely proportional to the consistency between the utterances they are confronted with and the standard language style they support. In other words, the smaller the surprisal, the more consistent the utterance is with their language style, and a larger surprisal indicates that the utterance deviates from their standard language style. Comparing surprisal under various conditions could not only assess the level of style-shifting but could also reveal valuable insights into the complex interplay of various factors that influence that change.

Based on the idea of surprisal, we consider two perspectives that are not typically considered in conventional metrics. The first perspective is the similarity of language models among individuals during a conversation. The second perspective is the relative change in language style compared with their usual conversation. These aspects are often under-emphasized in existing studies that rely on vocabulary comparisons between speakers within the conversation (Nasir et al., 2019; Kawano et al., 2020; Brugnoli et al., 2019). However, our approach takes both of these perspectives into consideration, enabling us to gain a deeper understanding of style-shifting by assessing stylistic alterations in relation to our own usual conversation.

**Perspective of similarity** We evaluate the similarity of language use between two speakers based on the surprisal of their language models for the target utterance $T$:

$$s_{s\bar{i}m} = |-\log p(T|v_{T \to S}) + \log p(T|v_{S \to T})| \quad (3)$$

Here, $v_{u_T \to u_S}$ denotes the case where the information of $u_T$ and $u_S$ is swapped and fed to the model. In other words, we evaluate whether $u_T$ speaks to $u_S$ or $u_S$ speaks to $u_T$ and whether they produce the same language. We excluded conditions $S$ and $v_I$ for calculating surprisal. This is because it may not be appropriate to use $S$ and $v_I$ as conditions in the same way when the roles of $u_S$ and $u_T$ are reversed. For instance, if $S$ is not a suitable utterance by $u_T$ in the first place, the surprisal will increase regardless of $T$. This similarity metric indirectly evaluates the similarity of their inherent language models. The score approaches zero when the two languages are similar (convergence), and has a large non-negative value when they are dissimilar (divergence).

**Perspective of style change** We also evaluate how the likelihood of an utterance changing under the influence of specific contextual factors, based on the speaker's usual utterances to the conversation partner. When specific contextual factors ($S$, $v_I$) are isolated from the input conditions of the language model, the change in the surprisal of $T$ can be evaluated:

$$s_{change} = |-\log p(T|v_{u_S \to u_T}) + \log p(T|S, v_{u_S \to u_T}, v_I)| \quad (4)$$

Here, $S_{change}$ takes 0 if the language style of $u_T$ is constant regardless of the given current context, and takes larger values if the language style of $u_T$ varies relative to their usual language style.

By combining our similarity and style change metrics, we can gain a deeper insight into the dynamics of style-shifting, including convergence and divergence in conversations.

In our style-shifting evaluation metrics, we evaluate the surprisal of a language model for an utterance as the average of all the surprisal for the words in that utterance. However, as in previous studies, we can also limit our evaluation to each marker word frequently occurring in the corpus. Furthermore, we can replace $S_{s\bar{i}m}$ with a simple similarity between utterances. Thus our method is compatible with the style-shifting analysis methods used in existing studies (Nenkova et al., 2008; Ireland et al., 2011).

## 4 Experimental Settings

This section describes dataset construction, training, evaluation criteria, and analysis framework.

### 4.1 Dataset Construction

We used Twitter data as conversations on social media to analyze style-shifting. From Twitter, to obtain the friends-followers (FF) relationships between users, reply pairs, and metadata associated with all users and tweets, we first selected five seed users: Japanese computer science researchers who are very active on Twitter, have many followers, and are well-known outside of social media. Then we obtained lists of their friends and followers. From the users in these lists, we obtained lists of friends and followers, representing second-degree FF relationships. Using the initial seed users and the relationship lists above, we built a social graph with users as nodes and "follow" relationships as directed edges. After that, we obtained all the tweets (from March 2018 to January 2023) from the node users in the graph. At this time, if a tweet refers to another tweet (reply, quote, etc.), we also obtained the source tweet.

For training the neural language model, we utilized 12,583,382 reply pairs tweeted between March 2018 and April 2022. We designated 860,701 reply pairs between users within the graph after April 2022 as the test set. Furthermore, the analysis of style-shifting was performed using conversations from this test set. In other words, we are assessing the style-shifting in conversations that were not part of the training data for our language models.

To verify the reliability of our metrics of style-shifting, we created an additional, smaller test set, which includes manual annotations. This dataset consists of 200 reply pairs randomly selected under the condition that each user within the pair has a record of more than 10 conversations and each user has conversed with other different users more than 5 times, from the test set. Two native evaluator for their language were asked to evaluate the style-shifting in the target utterance of each reply pair as follows:

- **Q1:** How different do you think this reply tweet is in style from the tweet to which it is replying? [1. Very different, 2. Somewhat different, 3. Rather different, 4. Neither, 5. Rather similar, 6. Somewhat similar, 7. Very similar]
- **Q2:** How different is this reply tweet from him/his usual style when they respond to the same user they are replying to? [1. Very different, 2. Somewhat different, 3. Rather different, 4. Neither similar nor different, 5. Rather similar, 6. Somewhat similar, 7. Very similar]
- **Q3:** How different is this reply tweet from him/his usual style when they respond to the other user they are replying to? [1. Very different, 2. Somewhat different, 3. Rather different, 4. Neither similar nor different, 5. Rather similar, 6. Somewhat similar, 7. Very similar]

The evaluators can refer not only to the target reply pairs themselves but also to their profiles, monologue tweets, and other conversations by the target user. They were asked to evaluate the style and tone, with word definitions referenced to the Japanese language dictionary as much as possible.

To ensure the reproducibility of our experiments, while we cannot release the raw dataset from Twitter, we will share the preprocessed dataset in accordance with Twitter's policies. Additionally, access to our code for both language model training and analysis will be provided[1].

---

[1] https://github.com/kwnsiy/style-shifting-eval

## 4.2 Training of Neural Language Models

We fine-tuned a Japanese GPT-2 model[2] comprised of approximately 336 million parameters. This model is constructed on a 12-layer transformer architecture with a hidden size of 768 and was pre-trained using the Japanese CC-100 and Japanese Wikipedia datasets. For the training (fine-tuning) process, we adopted a batch size of 64 and employed gradient accumulation steps of 2. We selected the AdamW optimizer with a learning rate set at $2 \times 10^{-4}$ and carried out the fine-tuning on four RTX3090Ti GPUs for up to 5 epochs. We set aside 5% of our training data as a validation set, choosing the model with the lowest perplexity (PPL) on this set for our evaluations. We also trained both node2vec and HITS scores using the same training dataset, representing features of users absent from the graph as zero or negative values. For further details, please refer to both the repository of pre-trained model and our own repository.

## 4.3 Evaluation of Neural Language Models

We used perplexity (PPL) to evaluate our personalized neural language models by using the evaluation dataset. We generated multiple models for comparison: our fully conditioned neural language model as well as models from which some or all the conditions were removed. For simplicity, we used the conditional probability of target $T$ predicted by our model when describing these models. The conditions for the language model include preceding utterance $S$, implicit user's social relations $v_{u_S \rightarrow u_T}$, and the potential social impacts of conversation $v_I$. Our hypothesis is that our conditioned personalized neural language model will better reproduce the language use of an individual, represented by a lower perplexity score.

## 4.4 Analysis of Style-Shifting

First, we investigated the correlation between the metrics $s_{sim}$ and $s_{change}$ proposed in Section 3 and the human evaluation in Section 4.1. As the baseline based on the word-based similarity, we introduced a score using Word Mover's Distance (WMD) (Kusner et al., 2015; Nasir et al., 2019; Kawano et al., 2020). We designed two intuitive baseline scores, $s_{sim}^{wmd}$ and $s_{change}^{wmd}$ corresponding to $s_{\bar{sim}}$ and $s_{change}$. $s_{sim}^{wmd}$ is calculated using a reply

---
[2]https://huggingface.co/rinna/japanese-gpt2-small

pair $(S, T)$ as follows:

$$s_{\bar{sim}}^{wmd} = \text{WMD}(S, T) \quad (5)$$

Here, WMD is a function that calculates the semantic distance between two utterances based on the word mover's distance. $s_{change}^{wmd}$ is calculated as follows, using the average of distances to reply utterances in another reply pairs with the same conversation partner from $T$ and the average of distances to reply utterances with different conversation partners from $T$.

$$s_{change,same}^{wmd} = \sum_{(S_{u_S}, T_{u_T}) \in C_{u_S, u_T}} \frac{\text{WMD}(T, T_{u_T})}{|C_{u_S, u_T}|} \quad (6)$$

$$s_{change,other}^{wmd} = \sum_{(S_{-u_S}, T_{u_T}) \in C_{-u_S, u_T}} \frac{\text{WMD}(T, T_{u_T})}{|C_{-u_S, u_T}|} \quad (7)$$

Here, $C_{u_S, u_T}$ is a set of other reply pairs by $u_S$ and $u_T$, and $-u_S$ is any user other than $u_S$.

We employed a logistic regression model to predict the binarized proposed style-shifting scores of reply pairs, using various non-linguistic factors described in Section 2.3 as input features. Specifically, we used the binarized $s_{\bar{sim}}$ and $s_{change}$ as the style-shifting scores. We use this model to identify important factors in style-shifting dynamics. To binarize such style-shifting scores, we set a threshold based on the 50th percentile of the reply pairs.

# 5 Experimental Results

## 5.1 Performance of Neural Language Models

Table 1 presents the perplexity scores of the neural language models on the test-set.

We evaluated the impacts of different conditions on perplexity. Perplexity can be improved if the following conditions are included: preceding utterance $S$, user's information $v_{u_T}$, social relationships information $v_{u_S \rightarrow u_T}$. Notably, providing $S$ as a condition leads to the most substantial improvement, suggesting that incorporating it allows a language model to make more context-specific predictions. Furthermore, including both $S$ and user information $v_{u_S \rightarrow u_T}$ leads to greater improvement in perplexity compared to using only $S$ and $v_{u_T}$. This suggests that the speaker's style is influenced not only by their own characteristics but also by those of their dialogue partner, indicating the importance of considering the interaction dynamics. Conversely, when specific linguistic context $S$ is ignored, the speaker's style is predominantly shaped by their own features rather than those of a dialogue partner.

Considering that information $v_I$ related to the tweet's social impact, perplexity is not improved by providing it as a single condition or in combination with other conditions. This suggests that the influence of $v_I$ on perplexity may be limited compared to other conditions or maybe over-fitted to conversations in training data. However, it is important that our style-shifting evaluation metric is designed to capture the variation in perplexity when different conditions are given to the language model.

Table 1: Test-set perplexity of neural language models

| Model objectives | Perplexity |
|---|---|
| $-\log p(T)$ w/o fine-tuning | 169.93 |
| $-\log p(T)$ | 15.20 |
| $-\log p(T\|v_{u_T})$ | 13.84 |
| $-\log p(T\|v_I)$ | 15.51 |
| $-\log p(T\|v_{u_T}, v_I)$ | 15.27 |
| $-\log p(T\|v_{u_S \to u_T})$ | 14.41 |
| $-\log p(T\|v_{u_S \to T}, v_I)$ | 14.62 |
| $-\log p(T\|S)$ w/o fine-tuning | 78.08 |
| $-\log p(T\|S)$ | 13.33 |
| $-\log p(T\|S, v_{u_T})$ | 12.63 |
| $-\log p(T\|S, v_I)$ | 13.72 |
| $-\log p(T\|S, v_{u_T}, v_I)$ | 13.86 |
| $-\log p(T\|S, v_{u_S \to u_T})$ | **12.59** |
| $-\log p(T\|S, v_{u_S \to u_T}, v_I)$ | 13.40 |

## 5.2 Analysis of Style-Shifting

### 5.2.1 Relationships with Human Evaluation

Table 2 shows the correlation between the average subjective evaluation results of two human evaluators and various style-shifting metrics, including conventional metrics. Here, $Q1$ indicates the similarity of styles between reply pairs, $Q2$ indicates the style changes compared to usual conversations with the same conversation partner, and $Q3$ indicates the style changes compared to usual conversations with different conversation partners. *, **, *** indicate that the $p$-value in the significance test of no correlation is less than 5%, 1%, and 0.1%, respectively.

The results show that while the conventional metrics did not correlate much with human evaluation, the similarity metric and style change metric we proposed did correlate to a certain extent with human evaluation. $s_{\bar{sim}}$ is designed to reflect the dissimilarity of styles between reply pairs, which showed a negative correlation with Q1, resulting as per our hypothesis. Similarly, $s_{change}$ is designed to reflect how much a speaker's utterances deviate from their usual style. It showed a negative correlation with both Q2 and Q3, again resulting

as per our hypothesis. The conventional metrics, in general, did not show a correlation with human evaluation results. This suggests the limitations of conventional metrics that evaluate only based on the similarity of words within observed utterances (Nasir et al., 2019; Kawano et al., 2020). Indeed, style is a very nuanced concept, and even if the words used between utterances are similar, it does not necessarily reflect the style.

Table 2: Correlation between human subjective evaluation results and proposed and conventional metrics

| Comparison | Correlation coefficient |
|---|---|
| Q1 vs. $s_{\bar{sim}}$ | $-0.22$ *** |
| Q2 vs. $s_{change}$ | $-0.13$ * |
| Q3 vs. $s_{change}$ | $-0.19$ ** |
| Q1 vs. $s_{\bar{sim}}^{wmd}$ | $-0.05$ |
| Q2 vs. $s_{change,same}^{wmd}$ | $0.09$ |
| Q3 vs. $s_{change,other}^{wmd}$ | $0.03$ |

However, it should be noted that the observed correlation with human judgments and our metrics, although statistically significant, is not particularly strong. This could be due to limitations in the performance of language models used for scoring and potential biases in the samples used for subjective evaluation. Moreover, another contributing factor is that while the subjective evaluation data was randomly sampled, style-shifting occurs relatively rarely in conversation, and the inter-rater reliability is not particularly high. We might need to focus on improving the sampling methodology for subjective evaluation.

### 5.2.2 Prediction of Style-Shifting

We examined the contribution of various non-linguistic factors of conversation to the prediction of style-shifting and evaluated the performance of predicting style-shifting in conversations. In Table 3, we present the prediction results for style-shifting using the test set, employing 5-fold cross-validation. The chance rate denotes the prediction performance if we were to consistently predict the same label. Our results derived from $s_{\bar{sim}}$ demonstrate that the logistic regression (LR) models, both with linear and second-order polynomial features, surpassed the chance rate, even without relying on text information. These findings underscore the superior performance in predicting style-shifting (either convergence or divergence). However, the prediction results based on $s_{change}$ exhibit only a marginal improvement compared to the chance rate. This highlights the necessity

for non-linear predictions that take into account both linguistic and non-linguistic factors, as well as their interplay.

Table 3: Performance of style-shifting prediction

| Prediction models ($s_{\bar{sim}}$) | Prec. | Recall. | F1 |
|---|---|---|---|
| Chance rate | 0.25 | 0.50 | 0.33 |
| LR | 0.59 | 0.59 | 0.59 |
| LR (Polynomial) | 0.61 | 0.61 | 0.61 |
| Prediction models ($s_{change}$) | Prec. | Recall. | F1 |
| Chance rate | 0.25 | 0.50 | 0.33 |
| LR | 0.53 | 0.52 | 0.49 |
| LR (Polynomial) | 0.54 | 0.53 | 0.52 |

Figure 2 shows the importance of the features of the logistic regression model based on $s_{\bar{sim}}$ and $s_{change}$, which has been trained on the entire dataset. Here, positive weights contribute to predicting different language uses (divergence) of the utterances of reply pairs, and negative weights contribute to predicting similar language uses (convergence). The asterisks indicate features where the result of the Wald test yields a $p$-value less than 5%.

**Prediction based on $s_{\bar{sim}}$** The results highlight the importance of implicit social meanings for prediction. Replies to authoritative interlocutors show linguistically different properties, which may be associated with maintaining uniqueness or expressing self-assertion. On the other hand, replies to interlocutors with high hubness show linguistically similar properties, which can be interpreted as reflecting adaptation to a style that promotes more effective communication or strengthens connections. If a reply $T$ gets many "replies", the reply shows linguistic similarity with the interlocutor. This suggests that the use of similar language may encourage communication involving the audience. Finally, the similarity of the implicit social meaning vectors of users $u_S$ and $u_T$ and the presence or absence of a follow relationship are associated with convergence. This suggests that mutual interests or similar attributes between users lead to a match in language style.

**Prediction based on $s_{change}$** The results highlight the engagement metrics associated with the reply pair $(S, T)$ significantly contribute to the prediction of style change. Additionally, each engagement metric reflects either style maintenance or changes differently, depending on its type. For instance, when both $S$ and $T$ garner many impressions, it suggests a wide audience observes their conversation. This broader audience might prompt both users to adjust their communication style to appeal to a more varied demographic. On the other hand, other metrics do not always yield such consistent results. This is due to actions such as replies, likes, and quotes reflecting different social meanings.

While the primary features we mentioned have non-zero coefficients as confirmed by the Wald test, they do not necessarily have strong coefficients. This offers a crucial insight for future research, emphasizing the need to identify additional features that may have a more potent impact on predicting style-shifting.

## 6 Related Work

There are existing methods for analyzing style-shifting that utilize language models, similar to our proposed method. Some studies construct unigram language models for each speaker in a conversation and compare differences in word likelihoods (Nenkova et al., 2008; Ireland et al., 2011). Another study evaluates whether one speaker's use of a marker increases the likelihood of its use by another (Danescu-Niculescu-Mizil et al., 2011). However, these methods often encounter issues with sparse training data due to the limited number of conversations involving the same speaker (Mizukami et al., 2016). As a result, they carefully select a small number of marker words that capture the syntactic features of conversation for training the language model. Moreover, they do not consider non-linguistic factors that could influence language style beyond speaker IDs. Our method addresses these issues by extending the existing methods that use unigram language models.

## 7 Conclusion

In this paper, we proposed a novel framework for analyzing style-shifting in social media conversations. Our framework captures changes in an individual's conversational style, leveraging surprisals predicted by a personalized neural language model. This model integrates not only the linguistic content of conversations but also non-linguistic factors such as social meanings. Compared to existing models, our personalized language model demonstrated superior performance in predicting an individual's language in a test set. Furthermore, our proposed metric for style-

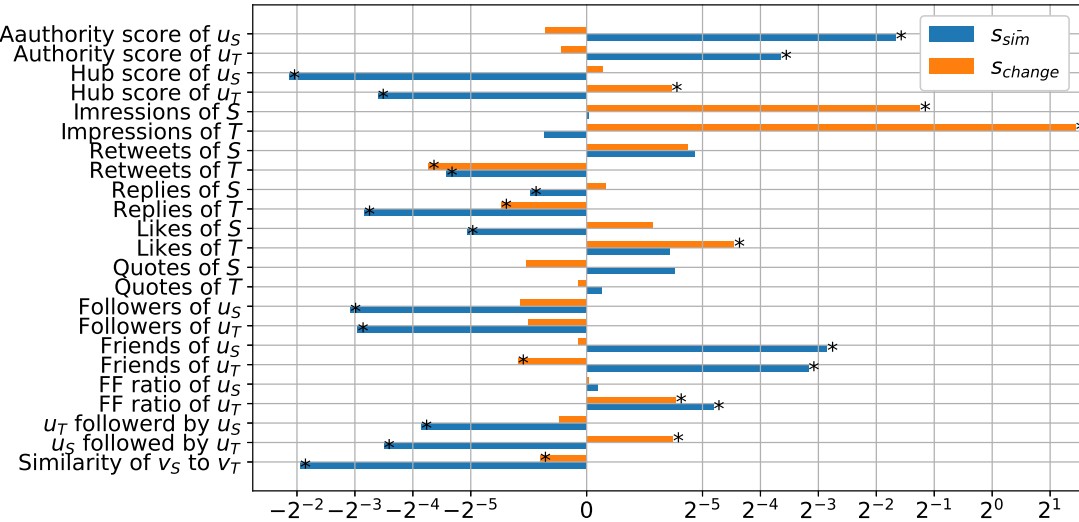

Figure 2: Importance features of style-shifting prediction based on $s_{\bar{sim}}$ and $s_{change}$

shifting analysis revealed correlations with various conversation factors and human evaluations of style-shifting. In conclusion, our proposed framework provides not only effective ways for analyzing style-shifting in conversations but also holds significant potential implications for enhancing dialogue systems. By applying the prediction models or heuristics that we have derived from our findings, dialogue systems may gain a more profound understanding of conversational dynamics and adjust effectively to varying conversational styles in different situations.

## Limitations

Our metrics for evaluating style-shifting rely on variations in surprisal as evaluated by personalized language models under various conditions. This estimate of surprisal is based on the assumption that the performance of the personalized language model is somewhat reliable. For instance, if a particular utterance is unusual under certain conditions and typical under others, there should be a variation in surprisal between them. Therefore, the reliability of evaluating style shifts in conversations of users with very limited training data, even when using few-shot learning, is diminished. Thus, there is a need to explore every means of training personalized neural language models.

Furthermore, our personalized language models incorporate various non-linguistic factors that could potentially influence style changes. However, the style is not explicitly handled within the model's latent representations. In other words,

while such non-linguistic factors are considered as input, their impact on the style of utterance generated by the model may not be explicitly captured. In our model, non-linguistic factors related to conversation participants and their utterances are incorporated as pre-calculated static features in the model training, but changes in interpersonal relationships or preferences over time are not considered. Our society changes over time, and individuals' preferences and linguistic behaviors change accordingly. It is important to consider such fluid nature in personalized embeddings.

## Acknowledgement

This work was supported by JSPS KAKENHI 22K17958 and JST ACT-X JPMJAX22A4.

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
