# OpenReview forum: "Analysis of Style-Shifting on Social Media: Using Neural Language Model Conditioned by Social Meanings"
_EMNLP/2023/Conference — EMNLP 2023 Findings_

### Official Review · Reviewer_UhTx · 2023-08-03

**Soundness:** 3

**Excitement:**

3: Ambivalent: It has merits (e.g., it reports state-of-the-art results, the idea is nice), but there are key weaknesses (e.g., it describes incremental work), and it can significantly benefit from another round of revision. However, I won't object to accepting it if my co-reviewers champion it.

**Paper Topic And Main Contributions:**

The paper proposes a method to evaluate style shifting in social media - accounting for both linguistic and non-linguistic factors. By incorporating non-linguistic factors into the first token of a GPT-based model and use the surprisal of the LM to determine the style change and similarity. The correlation with human annotations show better alignment of the scores to humans.

**Reasons To Accept:**

- Interesting analysis
- Use of linguistic and non-linguistic factors

**Reasons To Reject:**

- The paper is entirely statistical. For a task like this, it is important to show the linguistic nuance that is captured by the metrics.
- The choice of datasets: is 4 years sufficient period to study style shifts? what kind of style shifts happen in such a time? Without these answers, it is hard to appreciate what the model is capturing.

**Reproducibility:**

2: Would be hard pressed to reproduce the results. The contribution depends on data that are simply not available outside the author's institution or consortium; not enough details are provided.

**Reviewer Confidence:**

4: Quite sure. I tried to check the important points carefully. It's unlikely, though conceivable, that I missed something that should affect my ratings.

---

> ### Author Rebuttal · Authors · 2023-08-29
>
> Thank you for your insightful review and constructive feedback. We acknowledge the issues you have identified and plan to address them in the camera-ready version.
>
> Regarding the Analysis - To better describe how our proposed metric captures the nuances in style-shifting within conversations, we will include a case study in the paper. Specifically, we will show a comparison between the results of human evaluations and our style-shifting metric, across multiple conversation samples. This will help readers to better understand the areas in which our proposed style-shifting metric improves over prior methods.
>
> About the Dataset - We believe that our dataset, spanning four years, is robust and sufficient for evaluating the type of style-shifting we are exploring. In this paper, we define style-shifting as the degree to which a specific utterance deviates from an individual's typical conversational style; in other words, a local change of style, within a conversation. We operate under the assumption that this "typical style" is relatively stable over time (at least four years). Therefore, we are not considering very long-term changes in style, such as the transition from childhood to adulthood. We believe that exploring such temporal changes in style could be a promising direction for future research. To make these points clear, we will include further discussions in the camera-ready paper.
>
> Reproducibility - We understand the importance of reproducibility in research. To facilitate replication and verification by the community, we commit to releasing the code and dataset (anonymized as required). In the camera-ready paper, we will definitely include a valid download link for these resources.
>
> Your feedback is invaluable in improving the quality of our work. Thank you once again.

---

### Official Review · Reviewer_3agR · 2023-08-05

**Soundness:** 4

**Excitement:**

3: Ambivalent: It has merits (e.g., it reports state-of-the-art results, the idea is nice), but there are key weaknesses (e.g., it describes incremental work), and it can significantly benefit from another round of revision. However, I won't object to accepting it if my co-reviewers champion it.

**Missing References:**

The authors may cite more papers from ACL anthology by the keywords of "style variations".

**Paper Topic And Main Contributions:**

This study proposes a neural language model that identifies style shifts across the social media conversations by integrating user-level features. The authors collected a Twitter data of Japanese CS researchers. The study evaluated the model performance by the perplexity, proposed a new evaluation metrics by WMD, and conducted human evaluations. While the new proposed analysis method lacks of enough correlation with human judgement, the results show some promising directions. The authors also using the logistic regression to verify the validity of the proposed WMD method, the results show some merits of the approach.

**Questions For The Authors:**

1. Is the user set in the test as a subset of training users?
2. Is non-linguistic factor more helpful to improve the prediction performance?

**Reasons To Accept:**

1. The authors propose an interesting evaluation task.
2. The experiments show promising directions.

**Reasons To Reject:**

1. It is very hard to see validity of the proposed shift measurement from both human evaluation and automatic evaluation methods. For example, the correlation between the human results and the shifting values are low and mainly insignificant; and the prediction results by using the shifting values as features are close to a random guess (0.5).
2. There is no inference function of the personal (user) embedding. This may indicate that the experiments may need to keep the user set same across both training and test. We do not know if the proposed language model is generalizable.
3. Since the authors claimed their data contribution, we know very little about the data statistics.
4. Experimental settings such as implementation details and hyperparameters are not known.

**Reproducibility:**

4: Could mostly reproduce the results, but there may be some variation because of sample variance or minor variations in their interpretation of the protocol or method.

**Reviewer Confidence:**

4: Quite sure. I tried to check the important points carefully. It's unlikely, though conceivable, that I missed something that should affect my ratings.

**Typos Grammar Style And Presentation Improvements:**

The submission used `surprisals`, but it misused the concept multiple times as `surprise`.

---

> ### Author Rebuttal · Authors · 2023-08-29
>
> Thank you for your thoughtful review and constructive feedback. We acknowledge the issues you have pointed out and plan to address them in the camera-ready version.
>
> About WMD - To clarify, WMD serves as a baseline evaluation method for style-shifting in our study, rather than being the metric we propose. Our actual proposed metric relies on the concept of "surprisal," as derived from neural language models. We believe that your mention of WMD may have been a typo (probably you intend LM)  and wish to clarify this point to avoid any further confusion.
>
> Correlation with Human Judgment - As you pointed out, the observed correlation with human judgments, although statistically significant, is not particularly strong. This could be due to limitations in the performance of language models used for scoring and potential biases in the samples used for subjective evaluation. While the subjective evaluation data was randomly sampled, style-shifting occurs relatively rarely in conversation, and the inter-rater reliability (kappa=0.3) is not particularly high. We might need to work on improving the sampling methodology for subjective evaluation. Additionally, we note that our evaluation of style-shifting is conducted on test-set conversations (not a subset of training-set); these were not part of the data used to train language models. We will clarify these points in the camera-ready paper.
>
> Feature Importance - As you pointed out, the features we used in the prediction of style-shifting do not contribute "strongly" to the predictions. This highlights an important insight for future research: we need to identify additional features that could have a stronger effect in predicting style-shifting. However, it is worth noting that the contribution of the key features we mentioned in the results is not zero, as confirmed by Wald tests. We will clarify these points in the camera-ready paper.
>
> Importance of Non-linguistic Features - To emphasize the importance of non-linguistic features in predicting style-shifting, we will add a comparison of results between models that use only linguistic features and models that use both linguistic and non-linguistic features in Table 3. Additional experiments demonstrate improvement when both types of features are utilized.
>
> User Embeddings - We acknowledge your point that the set of user-specific embeddings is constructed from the training data. Hence, for new users appearing in the test set, we handle their vectors as zero vectors. Our subjective/objective evaluation only samples conversations that occur between users present in both the training and test sets. We are aware of this limitation and discuss it extensively in the "Limitations" section. Approximation techniques using the most similar user embeddings based on friends/followers lists could be a straightforward way to address this. We will add this discussion to the camera-ready paper.
>
> About the Dataset - Due to space constraints, only the size and collection period of the dataset were mentioned. We will add detailed statistics, including preprocessing methods, utterance lengths, and token counts, to the appendix in the camera-ready version.
>
> Training Settings - As stated in the paper, most hyperparameters are based on the pre-trained models. Details on batch size, optimizer, learning rate,  number of epochs,  computational environment, and main hyperparameters of GPT are already mentioned but will be further elaborated in the appendix in the camera-ready paper. We also intend to share the experimental code and dataset.
>
> Q1 - A set of users is created from the training data (mentioned above). However, the test set conversations are not a subset of the training data; they are non-overlapping and collected over different periods. Thus, we are tackling the challenging task of evaluating style-shifting in conversations not seen during the language model training. We are currently evaluating the impact of including the test set in the language model training. If we complete this in time, the findings will be included in the camera-ready paper.
>
> Q2 - Additional experiments show that using both language and non-verbal features improves the predictive performance of style-shifting.
>
> References - Thank you for your suggestion to include more papers related to "style variations." We will add the following studies into our paper. Additionally, we are continually investigating other potentially relevant literature to enrich the context of our study and will update our references accordingly.
>
> Kruijff-Korbayová, Ivana, et al. "The effect of dialogue system output style variation on users' evaluation judgments and input style." Proceedings of the fifth international natural language generation conference. 2008.
> Kang, Dongyeop, and Eduard Hovy. "Style is NOT a single variable: Case studies for cross-stylistic language understanding." Proceedings of the 59th Annual Meeting of the Association for Computational Linguistics and the 11th International Joint Conference on Natural Language Processing (Volume 1: Long Papers). 2021.
> Xu, Wei. "From Shakespeare to Twitter: What are language styles all about?." Proceedings of the Workshop on Stylistic Variation. 2017.
> Basile, Angelo, Albert Gatt, and Malvina Nissim. "You Write like You Eat: Stylistic Variation as a Predictor of Social Stratification." Proceedings of the 57th Annual Meeting of the Association for Computational Linguistics. 2019.
> Flekova, Lucie, Daniel Preoţiuc-Pietro, and Lyle Ungar. "Exploring stylistic variation with age and income on twitter." Proceedings of the 54th Annual Meeting of the Association for Computational Linguistics (Volume 2: Short Papers). 2016.
>
> English Issues - We apologize for the confusion regarding the terms "surprise" and "surprisal." We will carefully review the paper for typos and grammatical errors and plan to get additional checks by native English speakers.
>
> Thank you for helping us improve the quality of our work.

---

### Official Review · Reviewer_gqZW · 2023-08-06

**Soundness:** 3

**Excitement:**

3: Ambivalent: It has merits (e.g., it reports state-of-the-art results, the idea is nice), but there are key weaknesses (e.g., it describes incremental work), and it can significantly benefit from another round of revision. However, I won't object to accepting it if my co-reviewers champion it.

**Paper Topic And Main Contributions:**

This paper proposes a framework for evaluating style-shifting in social media conversations between connected individuals. Depending on the social media, the semantics of the connection may vary. The proposed framework captures changes in an individual's conversational style based on "surprisals" predicted by a personalized neural language model, which is the deviation from the LLM's own language. This is modeled by combining linguistic contents and non-linguistic factors such as social meanings, group membership, personal attributes, and individual beliefs via customized features mined from the social network. These factors are directly integrated in the model architecture proposed by the authors.

The personalized language model is claimed to outperform existing models in predicting an individual's language use in new conversations. It would seem that the entire gain is obtained by the effectiveness of the non-linguistic factors mined from the social network and combined in the language modeling. The framework introduces a novel metric for evaluating style-shifting based on surprisal, which measures the deviation of individual language from the overall model. The approach focuses on the surprisal experienced by a personalized neural language model, inversely proportional to the consistency between the model predictions and the individual's language style. The "surprisal" also directly depends on the hand constructed social network features.

The paper's overall contributions include the "surprisal" metric, the language model architecture, the analysis of correlations with user features on the social network.

**Reasons To Accept:**

1. Consideration of non-linguistic factors: By incorporating non-linguistic factors like social meanings and individual beliefs, the proposed framework accounts for the individual's relative viewpoint and provides a more comprehensive evaluation of style-shifting.

2. Comprehensive analysis: The research conducts a comprehensive analysis of style-shifting, revealing correlations between the proposed metric and various linguistic and non-linguistic conversation factors, as well as human evaluations of style-shifting. This analysis provides valuable insights into the complex interplay of factors influencing style-shifting.

3. Real-world application: The framework's applicability to social media conversations makes it relevant and practical for understanding conversation dynamics in online interactions. As social media communication continues to play a significant role in society, this work addresses a relevant and timely research topic.

**Reasons To Reject:**

1. Limited novelty - The proposed architecture is very similar to numerous existing models that attempt to combine linguistic and non linguistic features in language modeling. I would like the authors to present alternate architectures and provide an analysis or explanation for why some architectures may work better than others for this application.

2. Over-reliance on feature engineering: It is unclear to me how much the performance of the proposed approach is dependent on the features hand selected by the authors to control for non linguistic variations. The semantics of these features may considerably vary across platforms as well. Is the model robust if these features undergo shifts in values? Can we select a very different set of features on another platform and still observe meaningful results?

3. Limited reproducibility. The authors should ideally release code along with the paper via an anonymized repository to help reviewers validate the results.

**Reproducibility:**

3: Could reproduce the results with some difficulty. The settings of parameters are underspecified or subjectively determined; the training/evaluation data are not widely available.

**Reviewer Confidence:**

4: Quite sure. I tried to check the important points carefully. It's unlikely, though conceivable, that I missed something that should affect my ratings.

---

> ### Author Rebuttal · Authors · 2023-08-29
>
> We would like to express our gratitude for your review and constructive comments. We acknowledge the issues you pointed out and will implement the necessary revisions in the camera-ready version.
>
> Limited Novelty - We acknowledge that the architecture of our personalized language models is partially similar to existing language models. This point is already articulated in the related work section, where we have incorporated insights from previous studies into our models. To clarify further, our models are based on fine-tuning pre-trained models and extending their prefix embeddings without dramatically altering the underlying network structure or loss function. This approach is simple, but powerful, and easily replicable. Additionally, the main objective of this study is not to invent a new language model architecture but rather to focus on the application of personalized language models that incorporate various non-linguistic factors for the specific task of evaluating style-shifting. In the camera-ready paper, we will clarify this focus and discuss potential avenues for enhancing the architecture and loss function of language models. To provide readers with further insights into the unique features and capabilities of our language models, we will also present actual generated outputs. This will serve to emphasize the distinctive contributions of our study in leveraging non-linguistic factors, even if the architectural foundations are based on existing models.
>
> Overreliance on Feature Engineering - As you pointed out, our feature set could be problematic when evaluating style-shifting in other domains. However, as indicated in the title of the paper, our study aims to evaluate style-shifting in "social media." While the features we use are derived from Twitter, they are generally supported in other popular social media platforms (e.g., Facebook, Weibo, etc.). Hence, we believe they should not pose a significant barrier for other researchers wishing to validate our proposals on platforms other than Twitter.
>
> Reproducibility - We understand the importance of reproducibility in research. To facilitate replication and verification by the community, we commit to releasing the code and dataset (anonymized as required). In the camera-ready paper, we will definitely include a valid download link for these resources.
>
> Thank you for helping us improve the quality of our work.

---

### Meta-Review · Area_Chair_K31e · 2023-09-16

**Recommendation:** 4

**Metareview:**

This work proposes a method for the analysis of style-shifting on social media. The problem is interesting and relevant, the evaluation is robust, and the results are promising. The reviewers point out some issues for which the authors have provided a reasonable explanation, and they are eager to provide more context or fix them. Two main issues are related to reproducibility, for which authors will provide the code and data upon acceptance. The second one is related to the proposed method and the fact that it relies on hand-crafted features; I see this as an opportunity for further work.

Pro
* Interesting problem applied to a current topic and of interest to the community
* Robust evaluations
* Authors eager to address issues and release code and data
* Authors estimate the addition would be on time

Cons
* None

---

### Decision · Program_Chairs · 2023-10-07

**Decision:**

Accept-Findings

**Comment:**

This work proposes a method for the analysis of style-shifting on social media. The problem is interesting and relevant, the evaluation is robust, and the results are promising. The reviewers point out some issues for which the authors have provided a reasonable explanation, and they are eager to provide more context or fix them. Two main issues are related to reproducibility, for which authors will provide the code and data upon acceptance. The second one is related to the proposed method and the fact that it relies on hand-crafted features; I see this as an opportunity for further work.

Pro
* Interesting problem applied to a current topic and of interest to the community
* Robust evaluations
* Authors eager to address issues and release code and data
* Authors estimate the addition would be on time

Cons
* None